# Effects of Endosymbionts on the Nutritional Physiology and Biological Characteristics of Whitefly *Bemisia tabaci*

**DOI:** 10.3390/insects16070703

**Published:** 2025-07-09

**Authors:** Han Gao, Xiang-Jie Yin, Zhen-Huai Fan, Xiao-Hang Gu, Zheng-Qin Su, Bing-Rui Luo, Bao-Li Qiu, Li-He Zhang

**Affiliations:** Engineering Research Center of Biotechnology for Active Substances, Ministry of Education, Chongqing Normal University, Chongqing 401331, China; 2023110513028@stu.cqnu.edu.cn (H.G.); yinxiangjiecaptain@126.com (X.-J.Y.); 2024110513004@stu.cqnu.edu.cn (Z.-H.F.); xiaohang2202@outlook.com (X.-H.G.); su_zq34@163.com (Z.-Q.S.); 2024110513046@stu.cqnu.edu.cn (B.-R.L.)

**Keywords:** *Bemisia tabaci*, endosymbiont, host plant, amino acid, biological characteristics

## Abstract

This study concerns the nutritional contributions of symbionts conducive to the utilization of phloem sap, promoting insect growth and proliferation on host plants. In this study, we explored the relationships among host plants, symbionts, and insects from the perspective of nutrition. Our results showed that five populations of *Bemisia tabaci* MEAM1 that fed on different host plants contained the same species of symbiotic bacteria. Moreover, the biological characteristics and essential amino acid contents of different populations reared on different host plants were closely related to the titers of their primary symbiont, *Portiera*, which was further confirmed after antibiotic treatment. These results indicate that host plants can affect the titers of symbiotic bacteria, the synthesis of amino acids, and the biological characteristics of host insects.

## 1. Introduction

Arthropod insects are infected by various intracellular bacterial endosymbionts and form intimate symbiotic relationships with them [1]. These bacterial endosymbionts are usually transmitted from host to host via vertical transmission [2,3]. Bacterial endosymbionts are divided into two types: obligate (primary) endosymbionts and facultative (secondary) endosymbionts [4,5,6]. Primary endosymbionts form an obligatory mutualistic relationship with their hosts; i.e., they depend on each other for survival. There are many examples of this phenomenon, such as *Candidatus Buchnera aphidicola* in aphids, *Candidatus Carsonella ruddii* in psyllids, and *Candidatus Tremblaya princeps* in mealybugs, all of which produce nutrients for their insect hosts [7,8]. In contrast to primary endosymbionts, secondary endosymbionts are generally non-essential for host survival [8]. Hemipteran insects like whiteflies are phytophagous. Specifically, they feed on the sap of the phloem of carbohydrate-rich plants, but this sap lacks nitrogen-containing compounds, thereby causing an imbalance in their nutritional intake. The fundamental role of primary endosymbionts lies in their ability to biosynthesize critical nutrients absent from the host’s diet, particularly those nitrogen-based metabolites that the insect host cannot synthesize endogenously [9].

Whitefly *Bemisia tabaci* (Gennadius) (Hemiptera: Aleyrodidae) are notorious and devastating agricultural pests that have become major threats to global food security. They cause damage to crops by sucking the cell sap and excreting honeydew, thereby reducing the plants’ photosynthesis and growth, and they also transmit 38 plant diseases, leading to significant economic losses worldwide [10,11,12,13,14]. *B. tabaci* is a rapidly evolving species complex containing at least 40 cryptic species [15,16,17]. Different cryptic species of *B. tabaci* display significant differences in terms of their geographical distribution, host range, insecticide resistance, symbiotic bacteria composition, and capacity to spread viruses [18,19,20]. Among these species, the Middle East–Asia Minor 1 *B. tabaci* (MEAM1) is one of the most studied cryptic species on account of its significant capacity for invasion, the harm it poses, and its adaptation to more than 600 host plants in different habitats [13,17,18,19].

Previous studies have revealed one primary symbiont, namely, *Portiera*, and several secondary symbionts in *B. tabaci* populations, including *Arsenophonus* [21], *Cardinium* [22], *Hamiltonella* [23], *Rickettsia* [24], *Wolbachia* [25], *Fritschea* [26], and *Hemipteriphilus* [5,27,28]. Throughout their long-term co-evolutionary process, *B. tabaci* populations and their endosymbionts have established a close mutualistic relationship, being mutually essential regulators of development and adaptation [29]. For example, *Portiera* has been discovered in the specialized bacteriocytes located in the body cavity of *B. tabaci* and is closely related to nutrient synthesis, especially that of essential amino acids (EAAs) and carotenoids, which are missing from the phloem diet of host plants [30,31,32].

Within the complex plant–insect–endosymbiont relationship, most studies have focused on the plant–insect interaction or the insect–endosymbiont interaction, while few studies have investigated the complex relationships within the plant–insect–endosymbiont context [29]. Studies have shown that the egg production of *Rickettsia*-infected whiteflies is significantly higher than that of uninfected whiteflies, which also increases the number of females in the population of whiteflies. On the other hand, *Rickettsia*-infected whiteflies feed on host plants, which reduces plant defense and is beneficial to the insects feeding on the plants [33,34]. The endosymbionts in insects are affected by host plants, thereby affecting the growth, development, and survival of insects. Therefore, the importance of studying how host plants affect insect endosymbionts and how endosymbionts influence insects’ biological traits should be self-evident, being essential for furthering our understanding of the interactions among insects, endosymbionts, and plants.

In the current study, *B. tabaci* specimens with the same genetic background were cultured on six different host plants. We sought to investigate the following domains: (1) the direct effects of host plants on the species and titers of endosymbionts in MEAM1 populations reared on each of the tested host plants; (2) the total free amino acids (FAAs) content and the percentage of essential amino acids (EAAs) in different host plants and *B. tabaci* MEAM1 populations; and (3) the life-history traits of these *B. tabaci* populations as a consequence of their feeding and development on different host plants (including the body length, weight, and fecundity of these *B. tabaci*).

## 2. Materials and Methods

### 2.1. Plant Hosts

Five species of host plants, including cabbage (*Brassica alboglabra* Bailey), cotton (*Gossypium hirsutum* L.), tomato (*Lycopersicon esculentum* Mill.), tobacco (*Nicotiana tabacum* L.), and poinsettia (*Euphorbia pulcherrima* Willd. et Kl.), were used as experimental plants for whitefly *B. tabaci*. The seeds of cabbage, cotton, tomato, and tobacco were sown in plastic pots (12 cm diameter × 15 cm height) containing a soil–sand mixture (10% sand, 5% clay, and 85% peat), and poinsettia plants were purchased from the greenhouse in Dengtang Village, Baiyun District, Guangzhou. All the plants were kept in environmental chambers at 26 ± 1 °C and 70–85% RH.

### 2.2. Insects

Specimens of *B. tabaci* used in this study were initially acquired from pepper plants (*Capsicum frutescens*) in 2015 from the Engineering Technology Research Center of Pest Biocontrol, South China Agricultural University in Guangzhou. Populations were then reared on poinsettia under the following laboratory conditions: 26 ± 2 °C, 60% RH, and a photoperiod of 14:10 (L:D) h. Twenty pairs of parents were inoculated to the tested poinsettia, cabbage, cotton, tomato, and tobacco for natural reproduction, separately. After each population was propagated for one generation on each host plant, we replaced it with a new plant, and so on, until the *B. tabaci* on each host plant propagated to the 10th generation. Their colonies were monitored monthly by sequencing the mitochondrial *COI* gene used to maintain the purity, following the protocol used in Qiu et al. [35].

Populations treated with antibiotics: Among the 5 host plants, the nymphal development duration of *B. tabaci* feeding on cotton plants was the longest, so the cotton host plant was selected for the antibiotic treatment. Thus, the nymph was able to absorb more antibiotics, and the effect of the antibiotic treatment was stronger [36,37]. When the cotton grew to the 6–8 expanded leaf stage, the cotton was dug out from the plastic pots (12 cm diameter × 15 cm height), and we gently rinsed the soil on the root system with clean water to reduce the damage to the fibrous roots as far as possible. The cotton was cultured in Erlenmeyer flasks with a rifampicin concentration of 200 mg/L [38]. Twenty pairs of individuals from *B. tabaci* populations, reared in the laboratory and identified by mitochondrial *COI* gene sequencing, were released onto cotton plants to reproduce freely. New cotton plants were substituted when the initial ones wilted or grew excessively tall during the course of the experiment until the *B*. *tabaci* reached the 10th generation.

### 2.3. Endosymbionts Species and Their Infection in Different MEAM1 Populations Reared on Each Tested Host Plant

Thirty *B. tabaci* samples were randomly collected from each *B. tabaci* population, and *16S rDNA* and *wsp* genes were used to detect the endosymbionts (*Portiera*, *Hamiltonella*, *Rickettsia*, *Wolbachia*, *Fritschea*, *Cardinium*, and *Arsenophonus*) in these populations. The primer sequences and reaction conditions for each endosymbiont are shown in Table 1.

Total template DNA was extracted from single *B. tabaci* individuals that were collected from different populations following the method proposed by Liu et al. [4]. All the PCRs were performed in a 25 μL reaction volume, which included 1 μL of the template DNA lysate, 1 μL of each primer, 2.5 mM MgCl_2_, 200 mM for each dNTP, and 1 unit of DNA Taq polymerase (Invitrogen, Guangzhou, China) [27,42]. The amplified PCR products (5 μL) were electrophoresed in a 1% agarose gel containing Gold-View colorant in 0.5 × TBE for 20 min at 120 mA and then photographed on a UV transilluminator. When bands of the expected size were visible on the gels, 20 μL volumes of PCR products were sent to Beijing Genomics Institute (BGI), Guangzhou, China, for sequencing. The results were compared to known sequences using NCBI’s BLAST (ver. 2.16.0+) algorithm. The 30 *B. tabaci* adults were randomly divided into three repeats for testing. *Portiera aleyrodidarum* DNA from already-known positive *B. tabaci* individuals was used as a positive control, and ddH_2_O was used as a negative control to eliminate potential confounding variables.

The total numbers of tested adults infected by the seven endosymbionts for each different population were recorded in order to calculate their infection frequencies according to the following the formula: the number of individuals infected/the total number of individuals screened × 100%.

### 2.4. Quantitative PCR Detection of Endosymbionts in Different B. tabaci Populations Reared on Each Tested Host Plant

When the *B. tabaci* on each host plant propagates to the 10th generation, 10 pairs of newly emerged (<7 h) female and male *B. tabaci* adults were collected and inserted into 1.5 mL centrifuge tubes and store at −80 °C. Four replicates were collected for each *B. tabaci* population.

The DNA extracted from the adults of each *B. tabaci* population and antibiotic population were used as templates. Real-time fluorescence quantitative PCR was performed with the specific primers shown in Table 2. Three repeats were set for each amplification. The reaction system was as follows: 5 μL SYBR Premix Ex Taq II, 0.5 μL (10 μM) for forward or reverse primers, 2 μL ddH_2_O, and 2 μL DNA template. The cycling conditions were as follows: 5 min activation at 95 °C; 40 cycles of 30 s at 95 °C and 30 s at 55 °C; and finally, 20 s at 72 °C. A non-template negative control (ddH_2_O) was included for each primer set to check for primer dimers and contamination.

The mean normalized expression value of each endosymbiont was calculated by comparing the threshold cycle (Ct) of each target gene to that of the whitefly *β-actin* gene according to the 2^−ΔΔCt^ method [43].

**Table 2 insects-16-00703-t002:** Oligonucleotide primers used in quantitative PCR of endosymbionts in *Bemisia tabaci*.

Symbiont	Gene	Amplicon Size (bp)	Primer Sequence (5′-3′)	Reference
*Portiera*	*16S r DNA*	229	Port-F 5′-TAGTCCACGCTGTAAACG-3′ Port-R 5′-AGGCACCCTTCCATCT-3′	[44]
*Hamiltonella*	*16S r DNA*	243	Ham-F 5′-GCATCGAGTGAGCACAGTT-3′ Ham-R 5′-TATCCTCTCAGACCCGCTAA-3′	[45]
*Rickettsia*	*Citrate synthase* (*gltA*)	154	Glta-F 5′-CGGATTGCTTTACTTAC-3′ Glta-R 5′-AAATACGCCACCTCTA-3′	[44]
*β-actin*	*B. tabaci β-actin*	130	Actin-F 5′-TCTTCCAGCCATCCTTCTTG-3′, Actin-R 5′-CGGTGATTTCCTTCTGCATT-3′	[46]

### 2.5. Analysis of FAAs in Different B. tabaci Populations Reared on Each Tested Host Plant

Newly emerged *B. tabaci* adults (post-emergence age < 10 h) were randomly collected from each host-adapted MEAM1 populations (i.e., cabbage-, cotton-, tomato-, tobacco-, poinsettia-derived populations). A total of 20 mg was extracted from the MEAM1 populations in each treatment, kept in a 1.5 mL tube until fully homogenate, and shook for 2 min in a vortex shaker (QL-866, Qilinbeier, Nantong, China). The homogenates were subsequently centrifuged at 14,000 rpm for 10 min. A 1 mL aliquot of the resultant supernatant was mixed with an equal volume of n-hexane for the lipid phase separation through centrifugation at 10,000 rpm for 10 min. The supernatant was discarded, and 0.5 mL of the underlying aqueous phase was collected. This underlayer was mixed with an equal volume of 8% (*w*/*v*) sulfosalicylic acid and centrifuged at 10,000 rpm for 10 min to remove protein. A total of 0.5 mL of the resulting supernatant was concentrated to dryness. To redissolve the dried sample, 0.75 mL of sample buffer was used. Finally, the solution was filtered through a 0.45 μm membrane. The free amino acid in *B. tabaci* was analyzed using an amino acid analyzer (S433, sykam, Munich, Germany) [44].

The FAAs were determined using an automatic amino acid analyzer (L-8800, HITACHI, Tokyo, Japan). The percentage of the various amino acids (AAs) was calculated according to the following formula: the respective amino acid content/the total amino acid content × 100%. The total amino acid content was determined using an automatic amino acid analyzer (L-8800, HITACHI, Tokyo, Japan).

### 2.6. Body Size Measurement of B. tabaci from Different MEAM1 Populations Reared on Each Tested Host Plant

When the different populations of *B. tabaci* developed to the 10th generation, the newly emerged females and males were randomly collected. Their body size was measured from the top of the head to the end of the abdomen under a microscope. A total of 30 males and 30 females were measured in each population.

### 2.7. Weight Determination of Different B. tabaci Populations Reared on Each Tested Host Plant

We weighed an empty 1.5 mL centrifuge tube on a scale accurate to one-thousandth of a gram (FA2204B, Jingke, Shanghai, China) and recorded its mass. We then collected one thousand individual *B. tabaci* specimens from different host plant populations using a suction trap and gently transferred them into the centrifuge tube (wearing rubber gloves to prevent sweat from affecting the weighing results). We weighed the tubes containing the insects on the same scale accurate to one-thousandth of a gram. The difference in weight between the two measurements represents the weight of one thousand *B. tabaci* individuals. This process was repeated three times for each population.

### 2.8. Fecundity Determination of Different B. tabaci Populations Reared on Each Tested Host Plant

We selected one female and one male of the newly emerged *B. tabaci* and inoculated them onto the clean test plants, including cabbage, cotton, tomato, tobacco, and poinsettia. We recorded the fecundity (the number of eggs laid) of the females until their death, with 30 replicates per population.

### 2.9. Statistical Analyses

The infection comparisons of the endosymbionts in *B. tabaci* and their biological characteristics, such as their body length, the weight of a thousand individuals and their fecundity, the total AA content, and the percentage of EAAs and non-essential amino acids (NEAAs), were analyzed using one-way analysis of variance (ANOVA), and the significance analysis was performed via Duncan’s multiple range tests (*p* < 0.05). All statistical analyses were performed with SPSS (ver. 17.0, SPSS Inc., Chicago, IL, USA), and the graphs were drawn with SigmaPlot (ver. 10.0, Systat Software Inc., Chicago, IL, USA). Data were tested for normality (Shapiro–Wilks test) and homogeneity of variance (Levene’s test) before using parametric tests. Egg production, body lengths, body weight (per thousand), and amino acid molecules among the different treatments were arcsine-transformed wherever the data did not conform to a normal distribution. Two-way ANOVA was used to analyze the body lengths of male and female adults across the different treatments.

## 3. Results

### 3.1. Detection of Endosymbionts in B. tabaci Cryptic Species

PCR detection showed that *B. tabaci* was infected with at least one of the primary endosymbionts (*Portiera*) and two secondary endosymbionts (*Hamiltonella* and *Rickettsia*). No other endosymbionts were detected in this study (Figure 1).

When feeding on different host plants for 10 generations, the *Portiera* infection rate in different *B. tabaci* populations was 100%, while that of *Hamiltonella* in *B. tabaci* adults was 86.7%, 90.0%, 83.3%, 83.3%, and 73.3% across the five populations reared on poinsettia, cabbage, cotton, tomato, and tobacco, respectively. The infection rate of *Rickettsia* in *B. tabaci* adults was 80.0%, 83.3%, 86.7%, 80.0%, and 76.7% across the five populations reared on poinsettia, cabbage, cotton, tomato, and tobacco, respectively (Table 3).

### 3.2. Endosymbiont Titers in Different B. tabaci Populations Reared on Each Tested Host Plant

The relative titers of the primary endosymbiont, *Portiera*, and the secondary endosymbionts, *Hamiltonella* and *Rickettsia*, present in different *B. tabaci* populations were examined using qPCR. The results showed that the titer of *Portiera* was much higher than that of *Hamiltonella* and *Rickettsia*. In addition, the titers of *Portiera* and the two secondary endosymbionts varied across the different groups. For *Portiera*, the titers, from higher to lower, were as follows: cabbage > poinsettia > cotton > tomato > tobacco populations. Specifically, the highest titer was in the cabbage population, and the lowest was in the tobacco population. The titers for *Hamiltonella* and *Rickettsia*, from higher to lower, were as follows: poinsettia > cabbage > cotton > tomato> tobacco populations (Figure 2). Meanwhile, the titers of three endosymbionts in female adults were all higher than those in male adults.

### 3.3. Host Plant Effects on the Body Size of B. tabaci Adults

The body lengths of different male and female *B. tabaci* populations are shown in Figure 3. There were significant differences among the body lengths of females and males from the different *B. tabaci* populations (*p* < 0.05). The body lengths of the females and males, from longer to shorter, were as follows: cabbage > tomato > poinsettia > cotton > tobacco. The female body length of each population was significantly larger than that of the male (*p* < 0.05). The longest individual came from the cabbage population (929.23 μm), and the shortest came from tobacco populations (863.24 μm).

### 3.4. Host Plant Effects on the Weight of B. tabaci Adults

The body weight comparison (the average individual among one thousand individuals) with respect to different *B. tabaci* populations is shown in Figure 4. Similar to the body size, there were significant differences among poinsettia, cabbage, cotton, tomato, and tobacco populations (*p* < 0.05). The largest weight of *B. tabaci* came from cabbage populations, i.e., about 39.67 μg per individual on average, and the smallest came from the tobacco population, i.e., about 33.4 μg per individual on average. The order of weight for the different *B. tabaci* populations was as follows: cabbage > tomato > poinsettia > cotton > tobacco populations. The variation trend of *B. tabaci* body weight was consistent with that of the body length of *B. tabaci* across different populations.

### 3.5. Host Plant Effects on the Fecundity Different of B. tabaci Populations Reared on Each Tested Host Plant

The averaged female fecundity of the different *B. tabaci* populations is shown in Figure 5. There was no significant difference in the fecundity of *B. tabaci* across the poinsettia, cotton, and tomato populations. In contrast, the fecundity of *B. tabaci* from the cabbage population was significantly higher, and the fecundity of *B. tabaci* from the tobacco population was significantly lower with respect to the other *B. tabaci* populations (*p* < 0.05).

The variation trend of the fecundity was also consistent with that of the weight and the body length of *B. tabaci* across different populations.

### 3.6. The FAAs in the Different B. tabaci Populations Reared on Each Tested Host Plant

The total content of free amino acids (FAAs) and the percentage of essential amino acids (EAAs) in different MEAM1 populations reared on each tested host plant are shown in Figure 6. The results show that the total FAA contents in the MEAM1 populations reared on each tested host plant were significantly different (*p* < 0.05). The total FAA content was highest in the cabbage population, and the lowest was in the tobacco population. There was no significant difference in the FAA content between the cotton and tomato populations. The variation in the percentage of EAAs across the different populations of *B. tabaci* was correlated with the changes in total FAA content. Moreover, the trends in both EAAs and FAAs are in line with the trends observed in *Portiera* across diverse populations of *B. tabaci*.

### 3.7. Effect of Antibiotic Treatment on Portiera of Bemisia tabaci

After subjecting the insects to 10 generations of antibiotic treatment, the *Portiera* titers in both female and male *B. tabaci* specimens from both the antibiotic-treated group and the control group were measured. The findings revealed a significant reduction in the *Portiera* titers in both female and male *B. tabaci* specimens as a result of the antibiotic treatment (*p* < 0.05) (Figure 7).

### 3.8. Effect of Antibiotic Treatment on the FAAs of Bemisia tabaci

The total contents of FAAs in cotton and antibiotic-treated populations of *B. tabaci* are shown in Figure 8. The results show that the total FAA contents in the antibiotic-treated population of *B. tabaci* were significantly reduced. The proportion of EAA, such as via Arg, Ile, Met, His, Val, Leu, Trp, Phe, and Lys, was also significantly decreased (Figure 9).

The reduction trend with respect to the total FAAs and the proportion of the EAA components was consistent with the reduction in the *Portiera* titer in the antibiotic-treated *B. tabaci* population.

### 3.9. Effect of Antibiotic Treatment on the Body Length, Weight of Thousand Individuals, and Fecundity of Bemisia tabaci

The biological characteristics of cotton and antibiotic-treated cotton populations, including the body length, the average weight of one of 1000 whiteflies, and fecundity, were analyzed. It was found that these biological characteristics of the *B. tabaci* population were significantly affected by the antibiotic treatment. The body length, the average weight of one of 1000 whiteflies, and fecundity were all distinctly decreased in the antibiotic-treated population, which may be closely related to the decrease in the *Portiera* titer and EAA contents (Figure 10).

## 4. Discussion

Over the last two decades, the associations among plants, insects, and endosymbionts have become key areas of scientific investigation. Several fields have been surveyed for field populations of *B. tabaci* collected from different plant species in locations in China, revealing that at least three factors, i.e., biotype or genetic group, host plant, and geographical location, can affect the infection frequencies of the secondary symbionts in *B. tabaci* [8,47]. Other studies have also revealed that the species and infection of symbiotic bacteria are significantly affected by the insect host plants [48,49,50], as well as by the geographical environment [51]. For example, Wilkinson et al. (2001) detected significant differences in the density of endosymbiotic bacteria in aphids on three host plants [52]. Pan et al. (2013) determined the symbiotic bacteria contents of the *B. tabaci* populations on different host plants, which showed that host plants had significant impacts on the symbiotic bacteria density of *B. tabaci* Q [44]. In our study, we used *B. tabaci* as the experimental subject and demonstrated using quantitative PCR (qPCR) assays that the titers of the various endosymbionts (*Portiera, Hamiltonella*, and *Rickettsia*, etc.) of *B. tabaci* on different host plants (poinsettia, cabbage, cotton, tomato, and tobacco) were different, which affected both the synthesis of nutrients and the biological characteristics of *B. tabaci*. This phenomenon has not been reported elsewhere in the literature.

In this experiment, we used *Bemisia tabaci*, which has the same genetic background as the experimental material. All of these *B. tabaci* populations were infected with the primary endosymbionts *Portiera* and the secondary endosymbionts *Hamiltonella* and *Rickettsia*, although the levels of *Hamiltonella* and *Rickettsia* infection varied among the different populations reared on different host plants. Specifically, the *Portiera* titer in both the female and male cabbage populations of *B. tabaci* was found to be the highest, and significantly higher than in other populations. In contrast, the highest *Hamiltonella* and *Rickettsia* titers were observed in the poinsettia MEAM1 populations.

Previous studies have revealed that endosymbionts not only provide EAAs for insects but also provide lipids and vitamins through a series of biochemical actions [53,54]. Insufficient intake of any one of the EAAs, regardless of the total amount of the other AAs ingested, will prevent the normal growth and development of insects [55]. Thus, endosymbionts play an essential role in the regulation of insect nutrients. Primary symbionts are essential to host development and reproduction through their synthesis of the missing essential nutrients in the diet, such as EAAs and carotenoids [56]. In our study, the body lengths of females and males, as well as the weight and the fecundity of five populations of *B. tabaci*, were closely related to the titer of *Portiera*; thus, we speculate that the *Portiera* titer may affect the length, weight, and fecundity of the host by regulating the contents of EAAs in the host.

Our analysis of FAAs in the five populations of *B. tabaci* revealed that the changes in FAAs were generally consistent with the levels of *Portiera.* In order to further prove that *Portiera* can affect the biological characteristics of the *B. tabaci* host by regulating the content of EAAs in the *B. tabaci* host, we used antibiotics to reduce the content of *Portiera* so as to test whether the contents of EAAs and the biological characteristics change in *B. tabaci* with the change in *Portiera*. As previously mentioned, the EAA contents of *B. tabaci* decreased significantly after the antibiotic treatment, resulting in a significant decrease in the body length, body weight (per thousand), and fecundity of *B. tabaci*, accompanying the decrease in *Portiera.*

Our research results were consistent with the findings of Wilkinson et al. (2001) and Santos-Garcia et al. (2012), who showed that the total content of FAAs in cotton treated with antibiotics was significantly different from that of untreated cotton [52,57]. Santos-Garcia et al. (2012) support our contention that *Portiera* is able to synthesize the EAAs Thr and Trp as well as non-EAA Ser [57]. Previous studies have shown that AAs such as Ala, Val, Leu, Phe, and Lys are essential for the normal development of *B. tabaci*, affecting its biological characteristics such as its development duration, body weight, and emergence rate [58,59,60,61]. In our current study, Arg and Trp were seen to increase in the antibiotic-treated population, and the levels of EAAs, including Ile, Met, Met, His, Val, Leu, Phe, Thr, and Lys, were obviously decreased in the antibiotic-treated cotton population, which may be closely related to the decrease in *Portiera* and may, furthermore, explain the decrease in the body length, weight, and fecundity of *B. tabaci*.

In conclusion, our current study clearly indicates that host plants can greatly affect the titers of primary and secondary endosymbionts in *B. tabaci*. The decrease in the body size, body weight, and fecundity of *B. tabaci* is closely related to the titer decrease in *Portiera*, which plays an important role in FAA and EAA synthesis. Our findings will assist in strengthening our understanding of the interactions among endosymbionts, insects, and their host plants.

## Figures and Tables

**Figure 1 insects-16-00703-f001:**
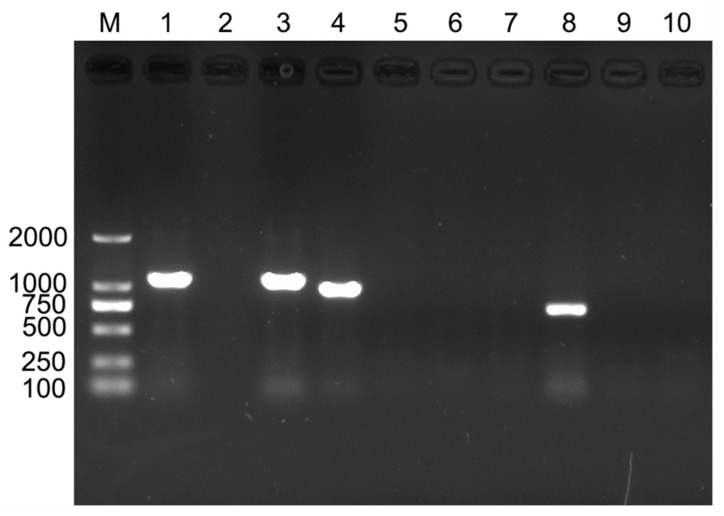
PCR detection of endosymbionts in *Bemisia tabaci*. M: DNA marker. Lanes 1–10 comprise the positive control (*Portiera*), negative control (ddH_2_O), *Portiera*, *Rickettsia*, *Wolbachia*, *Cardinium*, *Hemipteriphilus*, *Hamiltonella*, *Arsenophonus*, and *Fritschea*, respectively.

**Figure 2 insects-16-00703-f002:**
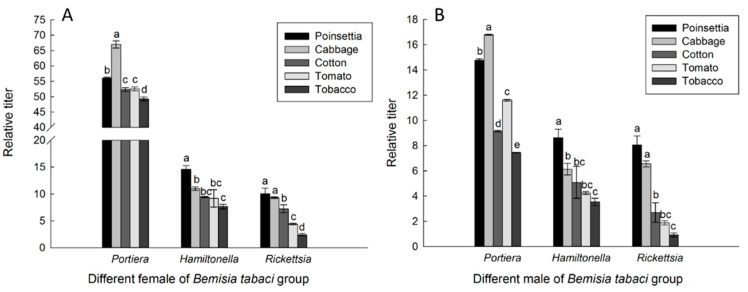
Relative titers of *Portiera*, *Hamiltonella*, and *Rickettsia* in the *B. tabaci* populations for the poinsettia, cabbage, cotton, tomato, and tobacco groups determined using quantitative PCR. (**A**) Different populations of female *B. tabaci*. (**B**) Different populations of male *B. tabaci*. The relative titers of the endosymbionts are given as the mean ± SD of three replicates. For each kind of endosymbiont, different letters above the bars indicate significant differences according to Duncan’s test (*p* < 0.05).

**Figure 3 insects-16-00703-f003:**
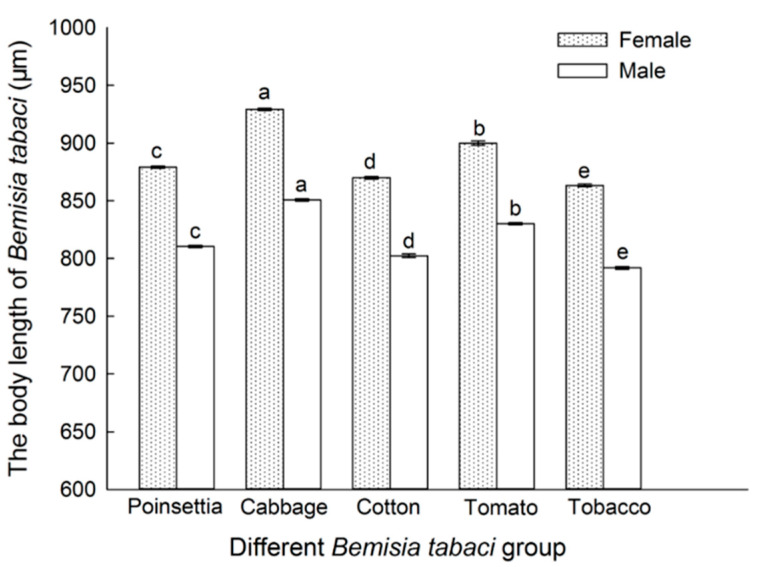
Body length comparison of *B. tabaci* adults in different populations. Values are represented as the mean ± SD. Different letters above the bars indicate significant differences among the five subclones according to Duncan’s test (*p* < 0.05).

**Figure 4 insects-16-00703-f004:**
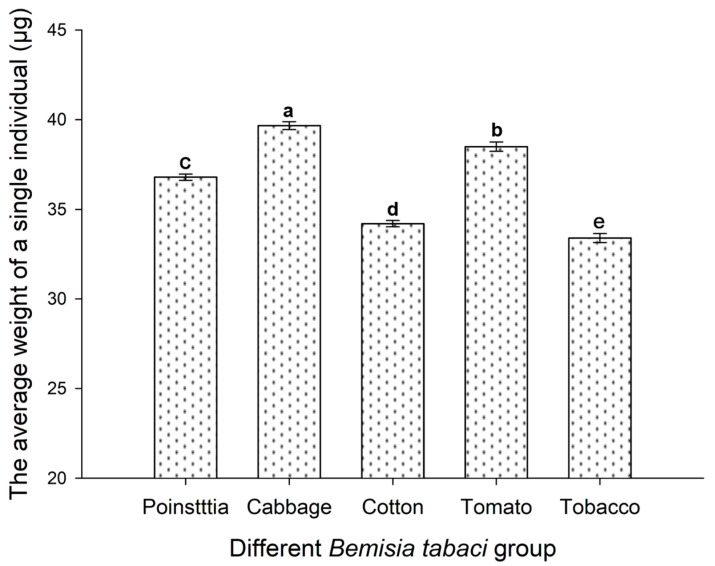
Weight comparison of *B. tabaci* across different populations. Values are represented as the mean ± SD. Different letters above the bars indicate significant differences across the five subclones according to Duncan’s test (*p* < 0.05).

**Figure 5 insects-16-00703-f005:**
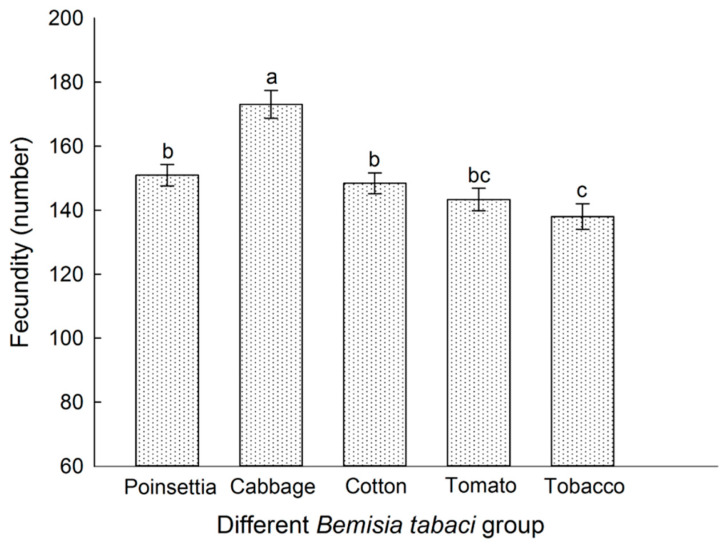
Fecundity comparison of *B. tabaci* in different subclones. Values are represented as the mean ± SD. Different letters above the bars indicate significant differences across the five populations according to Duncan’s test (*p* < 0.05).

**Figure 6 insects-16-00703-f006:**
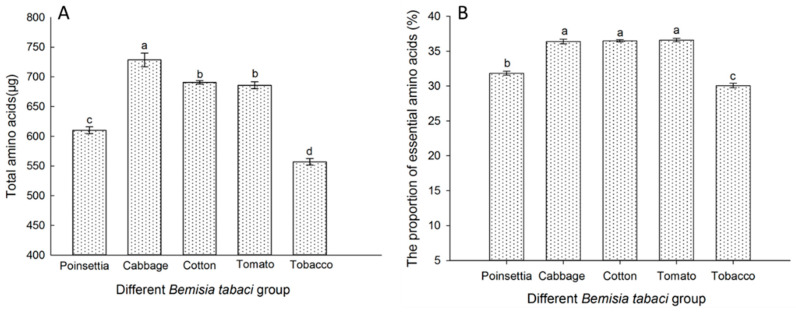
Total FAA content (**A**) and proportion of EAAs (**B**) in the five *B. tabaci* populations (poinsettia, cabbage, cotton, tomato, tobacco) analyzed using an amino acids analyzer. The values for the total AA contents are given as the mean ± SD of the three replicates. The data were analyzed using one-way analysis of variance. Different letters above the bars indicate significant differences across the five *B. tabaci* populations according to Duncan’s test (*p* < 0.05).

**Figure 7 insects-16-00703-f007:**
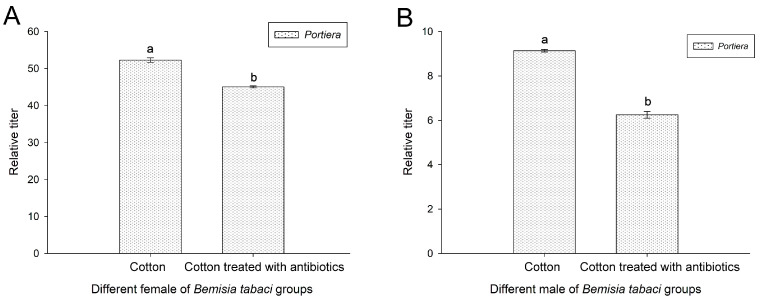
Relative titer comparison of *Portiera* in two *B. tabaci* populations (cotton and antibiotic-treated cotton populations): (**A**) female *B. tabaci* populations; (**B**) male of *B. tabaci* populations. Values for the relative titers of *Portiera* are given as the mean ± SD for the three replicates. The data were analyzed via one-way analysis of variance. Different letters above the bars indicate significant differences between the cotton and antibiotic-treated populations according to Duncan’s test (*p* < 0.05).

**Figure 8 insects-16-00703-f008:**
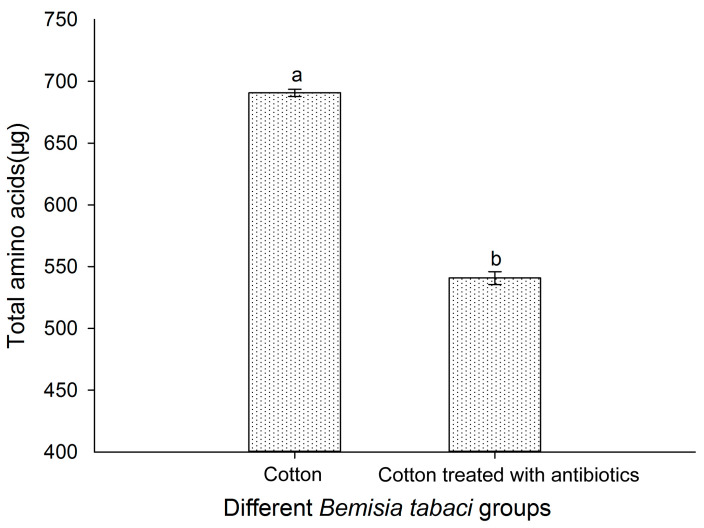
Total FAA contents in two *B. tabaci* populations (cotton and antibiotic-treated cotton populations). The values for the total FAA content of different *B. tabaci* are given as the mean ± SD of three replicates. The data were analyzed using one-way analysis of variance. Different letters above the bars indicate significant differences between the cotton and antibiotic-treated populations according to Duncan’s test (*p* < 0.05).

**Figure 9 insects-16-00703-f009:**
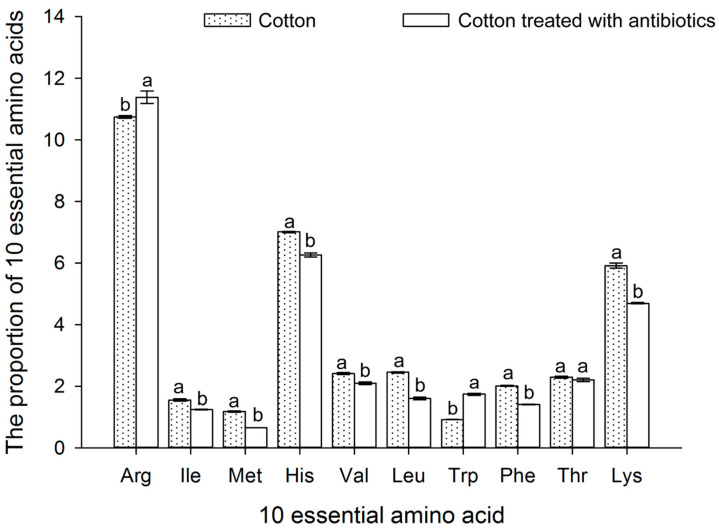
The proportions of 10 EAAs in the cotton and antibiotic-treated cotton *B. tabaci* populations. The values for the essential amino acid proportions in each *B. tabaci* population are given as the mean ± SD of three replicates. The data were analyzed using one-way analysis of variance. Different letters above the bars indicate significant differences across the 10 EAAs between the cotton and antibiotic-treated populations according to Duncan’s test (*p* < 0.05).

**Figure 10 insects-16-00703-f010:**
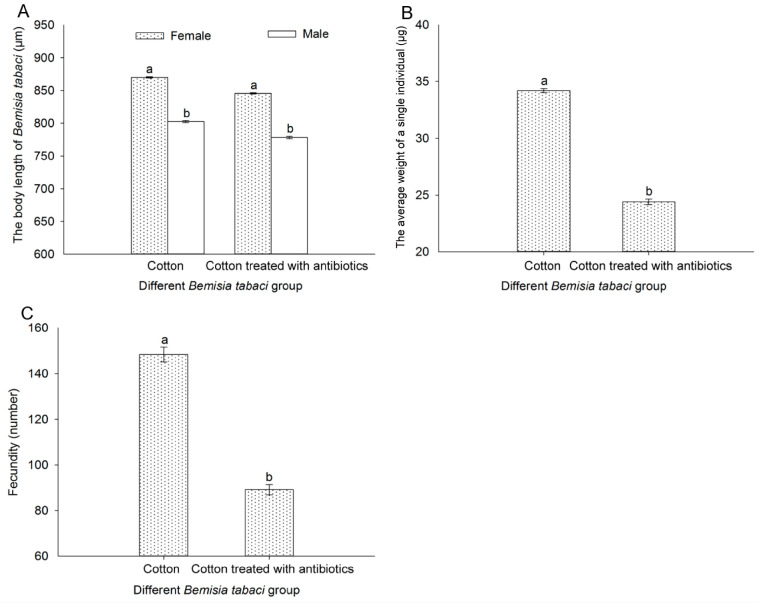
Comparison of the body length (**A**), weight (**B**), and fecundity (**C**) of *B. tabaci* populations on cotton and antibiotic-treated cotton plants. Data were analyzed using one-way analysis of variance. Values are represented as the mean ± SD. Different letters over the bars indicate significant differences between the groups at *p* < 0.05.

**Table 1 insects-16-00703-t001:** PCR detection primers and amplification conditions of seven endosymbionts of *Bemisia tabaci*.

Symbiont	Primer Sequence (5′-3′)	Annealing Temperature	Product Size (bp)	Reference
*Portiera*	28F: 5′-TGCAAGTCGAGCGGCATCAT-3′ 1098R: 5′-AAAGTTCCCGCCTTATGCGT-3′	60 °C	~1000	[23]
*Hamiltonella*	Ham-F: 5′-TGAGTAAAGTCTGGAATCTGG-3′ Ham-R: 5′-AGTTCAAGACCGCAACCTC-3′	58 °C	~700	[39]
*Rickettsia*	Rb-F: 5′-GCTCAGAACGAACGCTATC-3′ Rb-R: 5′-GAAGGAAAGCATCTCTGC-3′	58 °C	~900	[40]
*Wolbachia*	wsp-81F: 5′-TGGTCCAATAAGTGATGAAGAAAC-3′ wsp-691R: 5′-AAAAATTAAACGCTACTCCA-3′	55 °C	~600	[41]
*Fritchea*	U23F: 5′-GATGCCTTGGCATTGATAGGCGATGAAGGA-3′ 23SIGR5′-TGGCTCATCATGCAAAAGGCA-3′	55 °C	~600	[26]
*Cardinium*	CFB-F: 5′-GCGGTGTAAAATGAGCGTG-3′ CFB-R: 5′-ACCTMTTCTTAACTCAAGCCT-3′	58 °C	~400	[22]
*Arsenophonus*	Ars23S-15′-CGTTTGATGAATTCATAGTCAAA-3′ Ars23S-25′-GGTCCTCCAGTTAGTGTTACCCAAC-3′	52 °C	~600	[21]

**Table 3 insects-16-00703-t003:** Infection rates of different endosymbionts in the five populations.

Host Plant	*Portiera*	*Hamiltonella*	*Rickettsia*	*Wolbachia*	*Fritchea*	*Cardinium*	*Arsenophonus*
Poinsettia	100%	86.7%	80.0%	-	-	-	-
Cabbage	100%	90.0%	83.3%	-	-	-	-
Cotton	100%	83.3%	86.7%	-	-	-	-
Tomato	100%	83.3%	80.0%	-	-	-	-
Tobacco	100%	73.3%	76.7%	-	-	-	-

## Data Availability

The data that support the findings of this study are available in the Appendix A.

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
