# Peer review of "Effects of Endosymbionts on the Nutritional Physiology and Biological Characteristics of Whitefly Bemisia tabaci"

_insects, 2025, doi:10.3390/insects16070703_

Round 1
Reviewer 1 Report
Comments and Suggestions for Authors
This manuscript focuses on the interaction between the host plant, symbiotic bacteria and insects and investigates the influence of host plants on symbiont titers in Bemisia tabaci MEAM1, and examines how the endosymbiont Portiera affects nutritional physiology and biological characteristics. This work is interesting and has certain scientific value. However, I feel some issues and would like the author to revise the manuscript.
Q1, The line 229: the description of "Lanes 1-9" in Figure 1 is confusing (actually there are 10 lanes are listed), so it needs to be checked again and corrected.
Q2, "subclone" and "subgroup" are used interchangeably in this article, and "subgroup" is suggested to be unified to avoid confusion.
Q3, The “line 267: (per thousand). Does it refer to the weight of a thousand B. tabaci or does it mean something else? Please confirm in the text.
Q4, The title of Figure 7, Figure 8 and Figure 9 does not specify the control relationship of "antibiotic-treated group" (such as "cotton plant control group vs antibiotic treated group"), so it is suggested that the complete supplement should be made in the figure notes.
Q5, Some authors' names in the references are abbreviated (such as "B.-L.Q."), and some are not abbreviated (such as "Li-He Zhang"). Please make unified adjustments in accordance with the requirements of the journal.
Q6, There are certain grammatical errors in some parts of the text. Please pay attention to checking and correcting them.
Reviewer 2 Report
Comments and Suggestions for Authors
A brief summary
The aim of the paper is important, as understanding endosymbiont–host plant interactions in relation to insect pests is essential for developing species-specific management strategies. The authors have conducted valuable experiments that merit further recognition. The outcomes of this work will contribute to the whitefly research community by enhancing our understanding of the complex interactions between endosymbionts, host plants, and insect pests.
General comment
Overall, your manuscript requires both minor and major revisions. The minor corrections involve grammar, consistency, and clarity—particularly regarding terminology like “subgroups,” which should be replaced with more accurate terms such as “populations” or species names. Methods and results need more clarification, especially regarding the measured parameters. Major concerns relate to the purity of the MEAM1 population and the claim of shared genetic background, which are not sufficiently supported by your methods. To strengthen your study's credibility, you need to clarify your experimental design and address potential confounding factors due to population mixing. Improving the structure of your discussion to focus on the significance of your findings, rather than reiterating literature, will also enhance the manuscript's impact.
Article: One area of weakness is the method for generating a pure colony, which is not adequately described or controlled.
Abstract
- Line 21: The first mention of Bemisia tabaci should be written in full and include that it is a complex of cryptic species.
- Line 22: Avoid using "subgroups"; instead, specify the B. tabaci species used and mention that their host preference is host plant dependent.
- Line 23: “Same genetic background” needs clarification. What do you mean by “successive culture”?
- You have not mentioned the methods used to quantify endosymbiont titers and amino acid molecules. Please clarify.
- Line 24: Replace “amino acid species” with “amino acid molecules.” The word “species” is used for organisms, not molecules.
- Line 25: Specify what was measured in whitefly development and reproduction. Are you referring to fitness parameters?
- Line 26: B. tabaci is a cryptic species complex; saying "B. tabaci harbors" is vague. Specify the species name.
- Line 28: Clarify if you mean Portiera “content” or “titers.”
- Lines 28–39: Rewrite this section; it is unclear which statistical tests were used. Also, mention the significance level (p-values).
Introduction
- Line 51: Correct the sentence: “all of which have 50 been make nutrients for those insect hosts” to proper English.
- Lines 51–56: Rewrite this section due to grammar and formatting issues. Include that Sternorrhyncha is a suborder containing whiteflies, or mention Hemipteran insects like whiteflies.
- Lines 89–90: Clarify what is meant by "effect on species"—are you referring to prevalence? Specify what was measured.
- Lines 91–92: Clarify what you mean by “varieties.” What did you measure? Be specific about the parameters investigated.
Materials and Methods
- Line 101: You mentioned a soil-sand mixture (10% sand, 5% clay, 85% peat). Explain the reason behind this choice to inform the reader.
- Line 116: You stated that host choice was based on development time. Provide a scientific explanation. A longer development time does not necessarily mean better antibiotic absorption.
- Line 157: Avoid inconsistent terminology. Instead of “B. tabaci subclone” or “subgroup,” use: “MEAM1 populations reared on each tested host plant.”
- Line 171: Change “Twenty mg B. tabaci” to: “A total of 20 mg was extracted from the MEAM1 population in each treatment.”
- Line 173: Replace with: “the subsequent sample.”
- Lines 171–181: Rewrite for clarity.
- Line 182: Replace “subgroups” throughout the manuscript with: “different populations reared on different host plants.”
- Line 199: You did not describe how fecundity was measured. Was it based on the number of eggs laid? Clarify this.
- Line 206: Explain how the total amino acid content was calculated. Was it based on the sum across replicates per treatment?
Results
- Line 222: Replace “subgroups” with “populations,” e.g., “The five populations reared on poinsettia, cabbage, cotton, tomato, and tobacco, respectively.”
- Line 225: Omit “Figure 1” from the main text. Move to supplementary material if needed.
- Line 228: Clarify sample identification (e.g., from 1 to 9) so readers know which well corresponds to which bacterium.
Discussion
- Line 358: Use italics for B. tabaci throughout the manuscript.
- Lines 358–394: This part reads more like a literature review. Rewrite to include comparisons with your own findings and other studies.
Major corrections
Introduction
- Line 88: What tests did you perform to confirm that “B. tabaci MEAM1 had the same genetic background”? Briefly describe molecular and/or biological validation.
- Line 90: You mention “subgroups” but also claim “same genetic background.” This is confusing. Clarify whether your study used one genetic background or multiple populations.
Materials and Methods
- Lines 106–115: The claim about using a pure MEAM1 population is questionable. Rearing from 20 adult pairs does not ensure genetic purity. Starting with a single female (isofemale line) is the standard method for obtaining a genetically homogeneous population. Also, COI barcoding confirms only the mitochondrial lineage and does not detect nuclear introgression. If multiple cryptic species or MEAM1 subtypes were present initially, and no isolation was done over 10 generations, the resulting colony is likely a mix. Please clarify how you ensured colony purity.
Comments on the Quality of English Language
The manuscript requires careful revision for grammar, syntax, and clarity. Several sentences are awkward or unclear, and some terminology is used inconsistently or imprecisely (e.g., "subgroups" instead of "populations" or specific species names). The abstract and methods section, in particular, need clearer phrasing to accurately convey experimental design and data interpretation. Additionally, several areas contain formatting issues or improper scientific wording (e.g., “amino acid species” should be “amino acid molecules”).
A thorough language edit by a native English speaker or professional scientific editor is strongly recommended to improve overall readability and precision.
Author Response
Comments 1: Line 21: The first mention of Bemisia tabaci should be written in full and include that it is a complex of cryptic species.
Response 1: The reviewers are greatly appreciated for their valuable comments, this error has been corrected, and attention has been paid to the description of the latter half.
Comments 2: Avoid using "subgroups"; instead, specify the B. tabaci species used and mention that their host preference is host plant dependent.
Response 2: Thank you very much for your valuable comments from the reviewer, and we have corrected this error.
Comments 3: “Same genetic background” needs clarification. What do you mean by “successive culture”?
Response 3: Thank you very much for your comments. In our study, the same genetic background refers to the passage down from the same generation of MEAM1, and it is propagated under strict experimental conditions, and the population purity is tested every 1-2 months to ensure the strict consistency of the experimental materials. "successive culture" means continuous breeding and passage on the host plants to be tested. We have replaced it in the text to express our meaning more clearly.
Comments 4: You have not mentioned the methods used to quantify endosymbiont titers and amino acid molecules. Please clarify.
Response 4: We would like to thank the reviewer for the modification suggestions. We have added the method of detecting the titer of symbiotic bacteria and amino acids in the later section of Materials and Methods, and we have referred to the abstract format of some other articles in the journal. We are worried that the number of words in the abstract will be too large after adding the detection method, so we are not going to add the detection method in the abstract.
Comments 5: Line 24: Replace “amino acid species” with “amino acid molecules.” The word “species” is used for organisms, not molecules.
Response 5: Many thanks to the reviewers for their valuable comments, we have noted this issue and made corrections.
Comments 6: Line 25: Specify what was measured in whitefly development and reproduction. Are you referring to fitness parameters?
Response 6: Thank you very much for your valuable comments from the reviewers, and the measurement content has been supplemented.
Comments 7: Line 26: B. tabaci is a cryptic species complex; saying "B. tabaci harbors" is vague. Specify the species name.
Response 7: The reviewer's comments are greatly appreciated, and we have assigned species names to be described.
Comments 8: Clarify if you mean Portiera “content” or “titers.”
Response 8: The reviewer is greatly appreciated for his valuable comments, which have been described more accurately.
Comments 9: Lines 28–39: Rewrite this section; it is unclear which statistical tests were used. Also, mention the significance level (p-values).
Response 9: We are grateful to the reviewers for their suggestions for modification. We have put the content of statistical tests and significance levels in the Materials and Methods section, so we do not describe statistical methods in the abstract section.
Comments 10: Line 51: Correct the sentence: “all of which have 50 been make nutrients for those insect hosts” to proper English.
Response 10: Thank you very much for your valuable comments from the reviewers, and this part of the language has been revised.
Comments 11: Lines 51–56: Rewrite this section due to grammar and formatting issues. Include that Sternorrhyncha is a suborder containing whiteflies, or mention Hemipteran insects like whiteflies.
Response 11: Thank you very much for your valuable comments from the reviewer. We have rewritten this part and made a more perfect expression and description.
Comments 12: Lines 89–90: Clarify what is meant by "effect on species"—are you referring to prevalence? Specify what was measured.
Response 12: Thank you very much for your valuable comments. What we want to express in this part is the influence of host plants on the species and titer of the endosymbiotic fungi of B. tabaci, rather than the influence on the species.
Comments 13: Lines 91–92: Clarify what you mean by “varieties.” What did you measure? Be specific about the parameters investigated.
Response 13: Thank you very much for the suggestions made by the reviewers, and we have provided a supplementary description of this part of the measurement content.
Comments 14: Line 101: You mentioned a soil-sand mixture (10% sand, 5% clay, 85% peat). Explain the reason behind this choice to inform the reader.
Response 14: Thank you very much for your comments. This soil combination is a regular choice for our pot experiment. Cotton grows normally in this soil. Since we do not involve soil-related indicators and all treatments use the same type of soil, it will not affect our experimental results. (Fan, Z.-Y.; Liu, Y.; He, Z.-Q.; Wen, Q.; Chen, X.-Y.; Khan, M.M.; Osman, M.; Mandour, N.S.; Qiu, B.-L. Rickettsia infection benefits its whitefly hosts by manipulating their nutrition and defense. Insects 2022, 13, 1161. https://doi.org/10.3390/insects13121161)
Comments 15: Line 116: You stated that host choice was based on development time. Provide a scientific explanation. A longer development time does not necessarily mean better antibiotic absorption.
Response 15: Thank you very much for your valuable comments. Among these host plants, cotton has the longest development time of B. tabaci. The purpose of using antibiotics is to eliminate the endosymbiotic bacteria in B. tabaci, so the longer the development time of B. tabaci on cotton, the more antibiotics will be consumed. Whitefly has a shorter development time, so it will eat fewer antibiotics.
Comments 16: Line 157: Avoid inconsistent terminology. Instead of “B. tabaci subclone” or “subgroup,” use: “MEAM1 populations reared on each tested host plant.”
Response 16: Many thanks to the reviewer for their comments on the revisions, we have replaced these expressions.
Comments 17: Line 171: Change “Twenty mg B. tabaci” to: “A total of 20 mg was extracted from the MEAM1 population in each treatment.”
Response 17: Many thanks to the reviewer for the modification comments, we have corrected this statement.
Comments 18: Line 173: Replace with: “the subsequent sample.”
Response 18: We thank the reviewers for their suggestions for modification, which we have replaced.
Comments 19: Lines 171–181: Rewrite for clarity.
Response 19: Thank you very much for your valuable comments, and we have reformulated this part.
Comments 20: Line 182: Replace “subgroups” throughout the manuscript with: “different populations reared on different host plants.”
Response 20: Thank you very much for your valuable comments, which have been replaced with more accurate statements.
Comments 21: Line 199: You did not describe how fecundity was measured. Was it based on the number of eggs laid? Clarify this.
Response 21: We are grateful to the reviewers for their valuable comments, and we have provided a supplementary description of the measurement content.
Comments 22: Line 206: Explain how the total amino acid content was calculated. Was it based on the sum across replicates per treatment?
Response 22: Thank you very much for the valuable comments given by the reviewers, and the method of obtaining the total amount of amino acids is supplemented in this paper.
Comments 23: Line 222: Replace “subgroups” with “populations,” e.g., “The five populations reared on poinsettia, cabbage, cotton, tomato, and tobacco, respectively.”
Response 23: Thank you very much for your valuable suggestions from the reviewer, which we have corrected completely.
Comments 24: Line 225: Omit “Figure 1” from the main text. Move to supplementary material if needed.
Response 24: Thank you very much for your valuable comments. Our Figure 1 is to prove the different endosymbiotic bacteria in different host plants in B. tabaci, so it is better to prove the influence of host plants on the endosymbiotic bacteria species in B. tabaci. Therefore, we are prepared not to change the position of Figure 1.
Comments 25: Line 228: Clarify sample identification (e.g., from 1 to 9) so readers know which well corresponds to which bacterium.
Response 25: Thanks for the reviewer's valuable comments, we have made a more perfect supplementary explanation in the figure note.
Comments 26: Line 358: Use italics for B. tabaci throughout the manuscript.
Response 26: We are grateful to the reviewers for their valuable comments, and we have revised this error.
Comments 27: Lines 358–394: This part reads more like a literature review. Rewrite to include comparisons with your own findings and other studies.
Response 27: Thank you very much for your valuable comments from the reviewers. We have rewritten this part and added our own research findings and comparisons with other studies.
Comments 28: Line 88: What tests did you perform to confirm that “B. tabaci MEAM1 had the same genetic background”? Briefly describe molecular and/or biological validation.
Response 28: Thank you very much for your valuable comments. Because B. tabaci MEAM1 population in our laboratory is screened in advance, special attention will be paid to the protection of population purity during the feeding process, and the population purity will be tested every 1-2 months. The specific detection methods are as follows: Thirty adults were randomly selected from the MEAM1 population for PCR experiments. The PCR products were sent to Shanghai Sangon Biotechnology Company for sequencing, and the sequencing results were compared in NCBI to determine the purity of the population.
Comments 29: Line 90: You mention “subgroups” but also claim “same genetic background.” This is confusing. Clarify whether your study used one genetic background or multiple populations.
Response 29: Thank you very much for your valuable comments. We have unified this statement. In our study, we randomly collected 20 adult pairs from the MEAM1 population that had been screened in advance in the laboratory (ensuring population purity) and inoculated them to spawn on each tested host plant. Therefore, our study used populations of the same genetic background.
Comments 30: Lines 106–115: The claim about using a pure MEAM1 population is questionable. Rearing from 20 adult pairs does not ensure genetic purity. Starting with a single female (isofemale line) is the standard method for obtaining a genetically homogeneous population. Also, COI barcoding confirms only the mitochondrial lineage and does not detect nuclear introgression. If multiple cryptic species or MEAM1 subtypes were present initially, and no isolation was done over 10 generations, the resulting colony is likely a mix. Please clarify how you ensured colony purity.
Response 30: Many thanks to the reviewers for their valuable comments. Your screening method is very correct, because the population used in our study was screened in advance by the laboratory, and the screening method used was exactly the screening method proposed by the reviewer, starting with single-headed females. Because the population we used is the previous work of the laboratory, and this work is completely based on the previous population established, our research is directly starting from the step of having been screened and ensuring the purity of the population, and we regularly test the purity of the laboratory population to ensure the purity of the used population, Therefore, we did not describe the screening method in this paper.
Response to Comments on the Quality of English Language
Point 1: The manuscript requires careful revision for grammar, syntax, and clarity. Several sentences are awkward or unclear, and some terminology is used inconsistently or imprecisely (e.g., "subgroups" instead of "populations" or specific species names). The abstract and methods section, in particular, need clearer phrasing to accurately convey experimental design and data interpretation. Additionally, several areas contain formatting issues or improper scientific wording (e.g., “amino acid species” should be “amino acid molecules”).
Response 1: Thank you very much for the reviewer's suggestion on the quality of the English language of our manuscript. We have corrected the unclear expression, including that you mentioned "subgroup" instead of "population" or specific species name, or that "amino acid species" should be "amino acid molecules".
Point 2: A thorough language edit by a native English speaker or professional scientific editor is strongly recommended to improve overall readability and precision.
Response 2: The reviewers are greatly appreciated for their suggestions on the article, and we have invited professional scientific editors for language polishing.

Reviewer 3 Report
Comments and Suggestions for Authors
Dear Editor and Authors
The submitted manuscript presents a relevant and timely study that explores the relationship between Bemisia tabaci, its endosymbiotic bacteria, and various host plants. The authors have successfully addressed important questions related to host plant influence on symbiont titers, amino acid profiles, and key biological parameters of B. tabaci.
However, the main shortcoming of the manuscript lies in the graphical presentation of the results. Some of the figures are blurry. The different letters used to indicate statistical differences between bars should start from 'a' and proceed in order from left to right (a, b, c...) across all relevant figures. In Figure 1, the lines should be labeled. In Figure 7A and 7B, the two panels should be displayed at the same size. In Figure 9, the asterisks should be replaced with letters, as done in the other figures. In many figures, the label "Different Bemisia tabaci group" is present and should be reviewed. Most importantly, in Figure 3, the columns for male and female appear to be mislabeled, giving the impression that there is no difference between them—contrary to what is stated in the main text. Please reformat all figures, provide them in higher resolution, and carefully label statistical differences throughout.
Since the entire study is based on the Bemisia tabaci MEAM1 strain, it is not necessary to repeat "MEAM1" throughout the text. It would be sufficient to state it once clearly at the beginning and omit it in the rest of the manuscript for improved readability.
Please clarify the use of the terms amino acids (AA), essential amino acids (EAA), and free amino acids (FAA). It is not entirely clear whether the "essential" amino acids mentioned refer to those essential for humans or for Bemisia tabaci. Also, for consistency and clarity, it would be better to write out "amino acids" in full only at the first mention and then use the abbreviation "AA" throughout the rest of the text, as already done for FAA and EAA. Please revise this consistently throughout the manuscript.
Additionally, some minor suggestions are listed below for your consideration:
Lines 19–20: The statement that the mutualistic relationship between insects and their endosymbionts is established “on account of their long-term co-evolutionary processes” may be too strong unless supported. Consider a more neutral phrasing such as “Insects and their endosymbionts have a close mutualistic relationship.”
Lines 68–71: It would be beneficial to add a relevant reference to support the statements.
Lines 81-84: Lines 81–84: Please add a supporting reference.
Lines 89-92: Please consider rephrasing this sentence.
Lines 98-99: Please specify which biological traits are being referred to.
Lines 201-202: Please specify the sample size.
Line 208: Please specify the device or method used for accurate counting.
Line 207: Please specify the type and manufacturer of the scale used for weighing.
Figure 4: Please consider presenting the mass as the mean weight of a single individual rather than that of 1,000 individuals.
Line 329: Please move the instrument detail (L-8800, HITACHI, Japan) to the Materials and Methods section.
Lines 335–336: The text here appears to be repetitive, as similar content is already presented in lines 121–124. Consider revising to avoid redundancy.
Line 344: Could the authors please provide more details on how the relative titer comparison was performed? Please add it to the Materials and Methods section.
Line 387: The phrase “other previous” may be redundant; using just one of the two words should be sufficient.
Once the above comments are addressed—particularly the improvement of the graphical presentation of results—I would be pleased to recommend the manuscript for publication.
Kind regards,
Round 2
Reviewer 2 Report
Comments and Suggestions for Authors
The authors have provided a satisfactory and well-justified response to the concerns raised. They clarified that, in their study, the term "same genetic background" refers to the MEAM1 lineage maintained through successive generations under controlled experimental conditions. The population purity is regularly verified (every 1–2 months) to ensure consistent experimental material. They also explained that "successive culture" refers to the continuous rearing and passage of insects on the host plants being tested, and they have revised the wording in the manuscript for clarity. Based on these clarifications and revisions, I am satisfied with the response, and I consider the manuscript suitable for publication.
Author Response
Comments 1:The authors have provided a satisfactory and well-justified response to the concerns raised. They clarified that, in their study, the term "same genetic background" refers to the MEAM1 lineage maintained through successive generations under controlled experimental conditions. The population purity is regularly verified (every 1–2 months) to ensure consistent experimental material. They also explained that "successive culture" refers to the continuous rearing and passage of insects on the host plants being tested, and they have revised the wording in the manuscript for clarity. Based on these clarifications and revisions, I am satisfied with the response, and I consider the manuscript suitable for publication.
Response 1:Thank you very much to the reviewers for your valuable comments on the article. Your suggestions have made our article better. Once again, we express our sincere gratitude and best wishes to you.